

# Making inference from wildlife collision data: inferring predator absence from prey strikes

Peter Caley[1], Geoffrey R. Hosack[2] and Simon C. Barry[1]

[1] Data61, Commonwealth Scientific and Industrial Research Organisation, Canberra, Australian Capital Territory, Australia
[2] Data61, Commonwealth Scientific and Industrial Research Organisation, Hobart, Tasmania, Australia

## ABSTRACT

Wildlife collision data are ubiquitous, though challenging for making ecological inference due to typically irreducible uncertainty relating to the sampling process. We illustrate a new approach that is useful for generating inference from predator data arising from wildlife collisions. By simply conditioning on a second prey species sampled via the same collision process, and by using a biologically realistic numerical response functions, we can produce a coherent numerical response relationship between predator and prey. This relationship can then be used to make inference on the population size of the predator species, including the probability of extinction. The statistical conditioning enables us to account for unmeasured variation in factors influencing the runway strike incidence for individual airports and to enable valid comparisons. A practical application of the approach for testing hypotheses about the distribution and abundance of a predator species is illustrated using the hypothesized red fox incursion into Tasmania, Australia. We estimate that conditional on the numerical response between fox and lagomorph runway strikes on mainland Australia, the predictive probability of observing no runway strikes of foxes in Tasmania after observing 15 lagomorph strikes is 0.001. We conclude there is enough evidence to safely reject the null hypothesis that there is a widespread red fox population in Tasmania at a population density consistent with prey availability. The method is novel and has potential wider application.

# INTRODUCTION

Data arising from vehicle-wildlife collisions (termed "roadkill" for road vehicles, "wildlife strike" for aircraft, and hereafter "wildlife collision") are viewed as a potentially informative source of inference on trends in wildlife abundance. Doubts, however, remain on the appropriate methods for analysing such data, and whether useful information can be retrieved. Indeed, a key challenge when making inference from wildlife collision data is that it is essentially presence-only data. As such, we typically cannot resolve the incidence rate over the temporal and areal extent of the study from the raw data alone as we do not have information about the vehicle movement (traffic) factors (e.g., speed,

Corresponding author
Peter Caley, peter.caley@csiro.au

volume) and the abundance and behaviour patterns of the wildlife that leads to their interactions/collisions with vehicles. These factors clearly matter. For example, *Finder, Roseberry & Woolf (1999)* give an example of landscape factors influencing the collision rate of vehicles with wildlife, *D'Amico et al. (2015)* show that higher abundance leads to a higher collision rate, and *Hobday & Minstrell (2008)* show that vehicle speed influences the probability of a vehicle-wildlife collision.

Despite its limitations, wildlife collision data are sometimes the only source of information that a species is present in an area of interest (e.g., *Boles, Longmore & Thompson, 1994*; *Lubis, 2005*). Such data may be used as an alternative method of resighting to make inference on marked predator populations (e.g., *McClintock, Onorato & Martin, 2015*). Conversely, if sampling effort can be quantified, it should be possible to use a lack of wildlife collision data for a particular species to make inference on the probability of species presence/absence, and this is the key motivation of the new analysis approach that follows.

We note that where a numerical response function between a predator and its prey is known to exist, one can evaluate the expectation of the abundance of the predator species conditional on the prey species abundance. Furthermore, if numbers of predator and prey are sampled via the same observation/sampling process (e.g., vehicle collisions), then an absence of the predator can be used to infer extinction probabilities. The number of terrestrial carnivore species involved in incidents with aircraft is non-trivial.

In this paper we illustrate how simple conditioning on a second prey species, using biologically realistic numerical response functions, can produce a coherent numerical response relationship from wildlife collision data that can be used for practical inference on the population size of the predator species, including the probability of extinction. We apply the new approach to test hypotheses regarding the distribution and abundance of the red fox (*Vulpes vulpes*, Linnaeus 1758) population in Tasmania, Australia by analysing data on the aircraft collision incidence with foxes and their major prey species on airport runways.

## MATERIALS AND METHODS

### Case study background

It has been hypothesized on the basis of extraction of fox DNA from predator scats that the red fox is widespread (though rare) in Tasmania (*Sarre et al., 2013*). In contrast, an analysis that inferred the spatio-temporal distribution of foxes that could have generated the fox carcass discoveries suggests that the hypothesized widespread population is highly unlikely (*Caley, Ramsey & Barry, 2015*). The provenance of the fox carcass data on which this inference was based has been contested (*Marks et al., 2014*). Therefore, additional, independent inference is needed.

This paper considers a new, independent line of evidence to further the current debate, but before describing it we briefly explore the "widespread though rare" hypothesis as posed. We suggest that this hypothesis does not have a strong empirical basis but is a construct that enabled explanation of the differential sighting paradox: a widespread population as suggested by DNA evidence from scats, but too rare for all other observational methods to reliably detect (excluding unverified visual sightings). In contrast to the widespread though

low density hypothesis, empirical evidence of invading red fox populations in the presence of abundant prey populations (as is the case in Tasmania) predicts highly irruptive invasion dynamics with peak density typically achieved within 5-10 years of foxes first becoming apparent, and often sooner. This pattern is repeated across mainland Australia, for examples see Fig. 4 in *Jarman & Johnson (1977)*, Figs. 3 and 4 in *Short (1998)*, and data presented by *Abbott (2011)*. For the fox bounty data analysed by *Short (1998)*, the mean time from first bounty payment to the peak number, excluding pasture protection boards that only paid fox bounties in a single year, was 6.4 years ($n = 49$) (J Short, 2015, unpublished data). As would be expected during the irruptive phase when *per capita* resources are high, litter sizes were reported to be large compared with those back in England (*Abbott, 2011*). Finally, we note that the "widespread though rare" hypothesis is difficult to disprove directly through negative survey results, as the hypothesized population appears to be detectable only by the method that generated the hypothesis—a statistical modelling approach that makes inference on joint distribution of the observation process(es) and data is required (*Caley, Ramsey & Barry, 2015*).

It transpires that collisions between aircraft and foxes during take-off and landing are not uncommon in mainland Australia. This is not unsurprising, given that *Crain, Belant & DeVault (2015)* reported that the red fox made up 12% of reported carnivore incidents involving civilian aircraft in the USA, second only to coyotes (*Canis latrans*) on 40%, and the widespread nature of red foxes in Australia. If the fox population is indeed widespread in Tasmania, one would expect that the underlying collision rate should be equivalent, after correcting for influential covariates, to that observed on the Australian mainland where foxes are indeed widespread (*Van Dyck & Strahan, 2008*). The underlying ecological hypothesis is that a widespread predator population should occur at densities commensurate with the available prey population, as defined by the underlying numerical response relationship.

## Wildlife collision data

Significant resources are dedicated to the regulation and analysis of aviation safety. Reporting of wildlife strike statistics, including runway collisions, is mandatory under legislation (*Australian Government Federal Register of Legislation, 2003*). We sourced data on runway collisions at airports across Australia's states and territories from the Australian Transport Safety Bureau (ATSB) for the period 2002–2014 (*Australian Transport Safety Bureau, 2012*, and ATSB: http://data.atsb.gov.au/detaileddata).

The airstrike database for the period 2002–2014 was queried for species identified as fox, the individual records checked, and tallied for the Australian states and territories (Table 1). The Tasmanian fox population was inferred to have become widespread by, at latest, mid-way through this period in 2009 (*Sarre et al., 2013*). Under the null hypothesis of the population being introduced and establishing in the late 1990s, this would be expected given the previously mentioned high rates of population increase following successful establishment. Indeed, we argue that due to the presence of abundant prey, an invading fox population in Tasmania should increase at the maximum (intrinsic) rate.
**Table 1 Number of runway strikes for red foxes and lagomorphs (European hares & rabbits) recorded in Australian states and territories over the period 2002–2014.** Source: *Australian Transport Safety Bureau (2012)* and Australian Traffic Safety Bureau (http://data.atsb.gov.au/DetailedData).

| State | Fox | Lagomorph |
|---|---|---|
| Queensland | 7 | 45 |
| New South Wales | 10 | 34 |
| Australian Capital Territory | 3 | 3 |
| Victoria | 8 | 25 |
| South Australia | 10 | 21 |
| Western Australia | 4 | 9 |
| Northern Territory | 0 | 0 |
| Tasmania | 0 | 15 |

There are about 50 airfields in Tasmania, of which *c*. 27 are spread through eastern Tasmania within the convex hull incorporating the fox DNA evidence underpinning the widespread hypothesis (Fig. 1). Of these airfields, wildlife strike data are recorded for Bridport, Devonport, Hobart, Launceston and Wynyard—all lie within moderate to high predicted fox occupancy under the widespread hypothesis (*Sarre et al., 2013*), and many in close proximity to evidence locations (Fig. 1).

## Analysis

Logically, vehicle movements and transport corridor environment needs to be taken into account when inferring the expected number of wildlife collisions, though these data are typically not available. Indeed, for our case study, compiling aircraft movement data and attributes of Australian airfields is an onerous task. There are *c.* 1,650 airfields covered by air services Australia, and even if traffic movement data were collected for all airfields there is still uncertainty about the resident population of species of interest and their access to the airport runways (perimeter fencing integrity etc.). Thus collection of more detailed information may not resolve the uncertainty around interpretation of the raw fox runway strike counts, and for wildlife collision data more generally.

To address these issues we present an alternative method of analysis. European hares (*Lepus europaeus* Pallas 1778) and European rabbits (*Oryctolagus cuniculus* Linnaeus 1758), collectively termed lagomorphs, are consistently a significant component of fox diet (*Saunders et al., 1995*), and there is expected to be a numerical response relationship between the two (*Pech et al., 1992*). The distribution of foxes on mainland Australian is strongly influenced by the distribution and abundance of lagomorphs where they occur concurrently over the vast majority of their ranges. The numerical response of foxes to rabbits is strongly non-linear and largely concave (*Pech et al., 1992*).

We propose using the runway strike rate of lagomorphs by aircraft as a proxy for both the hazard rate posed by aircraft movements as well as reflecting the productivity of the airfield environment and available access of the runway for small mammals (in this case foxes and lagomorphs). We aggregate the strike data for each state in Australia and analyse the number of fox runway strikes conditional on the number of lagomorph runway strikes.

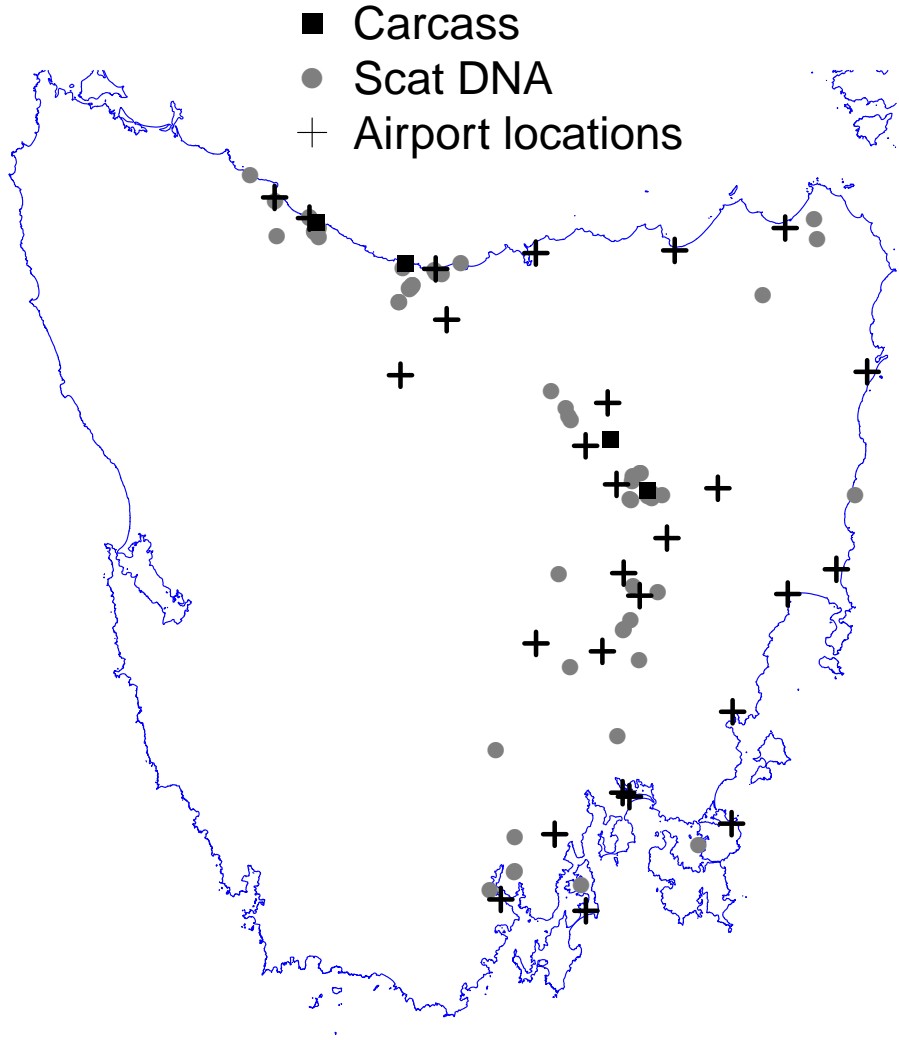

**Figure 1 Map of Tasmania (excluding islands in Bass Strait) showing the locations of fox scat-DNA and carcass evidence underpinning the widespread hypothesis of** *Sarre et al. (2013)* **and the locations of active airports from which runway strike data were used.** Note that airports not considered are not marked.

A degree of aggregation is necessary to overcome computational problems that would arise from the sparseness of these data. Other spatial resolutions would be possible, though the state-based approach had the advantage of the data already being classified by state. It is possible that the distribution of foxes and rabbits is not constant across a state. Indeed, in the Northern Territory the northernmost limit for rabbits and foxes is around Tennant Creek. The variable runway strike rate is accommodated for in the analysis by noting that the sum of multiple independent Poisson processes is again distributed as Poisson (see model specification below), with the intensity parameter the sum of the individual intensities (in this case the individual airport strike rates). The key assumption is that the numerical response function holds across space.

We note that the statistical conditioning enables us to account for unmeasured variation in factors influencing the runway strike incidence for individual airports, enabling valid comparisons—an important point that was not immediately apparent to other colleagues when first introduced to this approach.

We use a Holling type III numerical response (following *Pech et al., 1992*) to accommodate the possibility that foxes may be particularly reluctant to venture onto airstrips at low rabbit densities (analogous to prey-switching behaviour).

In statistical terms, the observed number of runway strikes ($y$) model is distributed as Poisson with underlying rate parameter ($\mu$):

$$y_i \sim Poisson(\mu_i). \tag{1}$$

The Holling type III numerical response relating the mean fox strike rate ($\mu_i$) to lagomorph runway strikes ($x_i$) is:

$$\mu_i = \frac{\beta_0 x_i^2}{\beta_1^2 + x_i^2}. \tag{2}$$

The subscript $i$ denotes the $i$th state or territory, and $\beta_0$ and $\beta_1$ are regression coefficients. $\beta_0$ is the asymptotic (maximum) prey abundance. Having $\beta_1$ squared facilitates its interpretation as the prey density at which the predator density is half of the maximum (see Results). Note that the numerical response relationship considered here relates predator abundance to prey abundance (*Caughley & Sinclair, 1994*, p. 172), as compared to predator population growth rate in relation to prey abundance (*Sibly & Hone, 2002*).

The model was fitted using standard Markov Chain Monte Carlo (MCMC) techniques with normal and positive zero-truncated normal priors for $\beta_0$ and $\beta_1$ with mean 0 and standard deviation 1,000, respectively. Three parallel MCMC chains were run with different starting values for $\beta_0$ and $\beta_1$. Convergence of the sampler was assessed visually using trace plots and Gelman and Rubin's convergence diagnostic (*Gelman & Rubin, 1992*).

The model was fitted to the mainland data, with the Tasmanian fox runway data point treated as missing. This approach enabled sampling a posterior predictive distribution for the Tasmanian fox runway strike incidence rate (Poisson intensity parameter for the survey period). This posterior distribution for the intensity parameter was then used in two ways. First, to generate the posterior distribution for the number of runway strikes of foxes in Tasmanian for the 2002–2014 period, conditional on the observed number of lagomorph runway strikes, including 95% and 99% prediction intervals. Second, and more to the point, we can estimate the probability of observing no fox runway strikes in Tasmania, conditional on the observed number of lagomorph strikes in Tasmania in conjunction with the numerical response fitted to mainland runway strike data. We also calculated credibility intervals across the range of lagomorph runway strikes observed across all states and territories. The model was fitted using OpenBUGS within the R software environment (*R Core Team, 2014*) with the "R2OpenBugs" R package (*Sturtz, Ligges & Gelman, 2010*).

## RESULTS

There was a strong relationship between the number of fox strikes and lagomorph strikes (Fig. 2). The 95% credibility interval for $\beta_0$ was greater than zero (95% CI [6.1–15.1]),

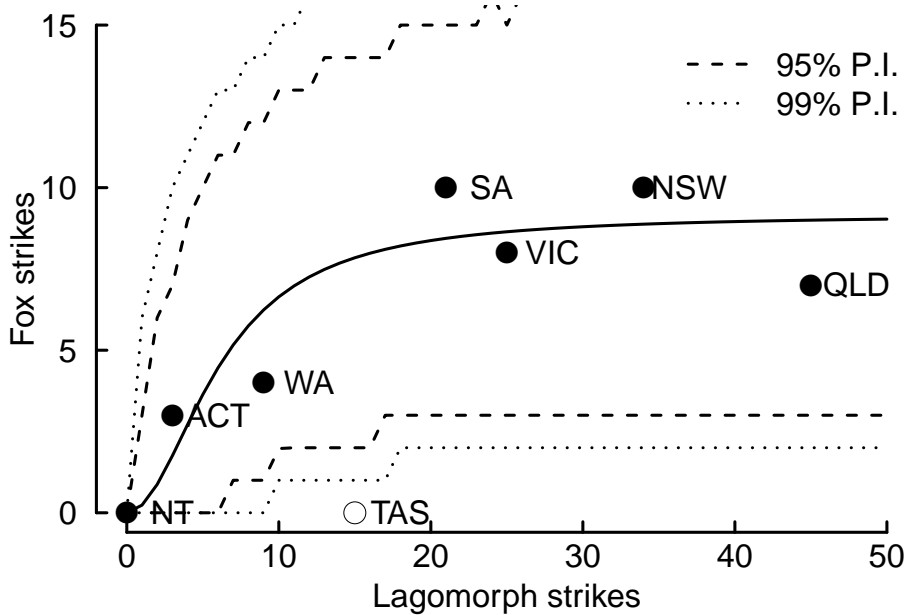

**Figure 2** **Numerical response relationship between the number of fox runway strikes versus lago-morph (hare or rabbit) runway strikes over the period 2002–2014 for Australian states and Territories.** Labels are: "NT"–Northern Territory; "QLD"–Queensland, "NSW"–New South Wales, "SA"–South Australia; "WA"–West Australia; "VIC"–Victoria; "TAS"–Tasmania. Source: *Australian Transport Safety Bureau (2012)*. Solid line is Holling type III numerical response model fitted to data with Tasmanian data point (open circle) omitted. Dotted and dashed lines are 95% and 99% prediction intervals (P.I.) for observations conditioned on the mainland data only.

indicating a significant, positive effect of lagomorph runway strike rate on fox runway strike rate. The *a priori* expectation of non-linearity of the numerical response is confirmed by the small estimate for $\beta_1$ (median = 6.2, 95% CI [1.5–17.5]), which is the number of incident lagomorph runway strikes at which half the maximum number (asymptote) of fox runway strikes would occur.

Under the hypothesis of a widespread fox population in Tasmania, on the basis of observing 15 lagomorph runway strikes over the 2002–2014 period, Tasmania is predicted to have a mean runway strike incidence of 7.6 period$^{-1}$ (95% credibility interval for the mean 5.3–10.2 period$^{-1}$) (Fig. 2). In terms of observations, the lower 95% and 99% posterior prediction interval bounds are 2 and 1 fox runway strikes for Tasmania over the 2002–2014 period, respectively (Fig. 2).

The probability of observing zero fox runway strikes in Tasmania, conditional on the fitted numerical response and the number of lagomorph strikes in Tasmania, was 0.001. That is, the probability of a widespread fox population existing in Tasmania whose abundance is consistent with the observed prey population is very low. That is, we reject the null hypothesis that there is an extant fox population in Tasmania that is ecologically consistent with the available prey population.

## DISCUSSION

Our study is the first that we are aware of to use airport runway wildlife strike data to make inference on the relative abundance of an invasive species. It does so through combining ecological theory (in this case a numerical response function) with an appropriate statistical model. This novel approach neatly circumvents, through the process of statistical conditioning, the difficulties of measuring the airport-specific risk of runway strike. Furthermore the Bayesian approach facilitates assigning probabilities to observed outcomes given the observed data and chosen model. There are undoubtedly additional applications of this approach to wildlife strike data. For example, *Crain, Belant & DeVault (2015)* document 1,016 incidents with civilian aircraft in the United States involving at least 16 species of carnivore. Clearly the method is equally applicable to roadkill data.

We have considered a new, independent line of evidence to further the current debate as to the distribution and abundance of foxes in Tasmanian. The inference from our analysis of independent data strengthens the case against their having ever been a widespread fox population in Tasmania that was ecologically consistent with available prey populations, or consistent with the detection probabilities of known observation methods. Accepting the alternative hypothesis (that there is not an ecologically consistent, detectable fox population in Tasmania) is not equivalent to inferring there is no fox population in Tasmania. Note, however, that both *Caley & Barry (2014)* and *Caley, Ramsey & Barry (2015)* infer that extinction is the most likely outcome. Our results are consistent with the rarity of road killed foxes in Tasmania in comparison to mainland Australia where they numerous—there have only been three fox carcasses found on Tasmanian roads and one produced by a hunter that were considered credible by Tasmanian authorities. We note, however, that the provenance of these carcasses is the subject of a polemical debate, and the deliberate hoaxing for some, or all of these carcasses cannot be definitively ruled out.

Looking to the future, if no further credible evidence of free-living foxes is found in Tasmania, then exactly how widespread and abundant foxes have been in Tasmania will undoubtedly be subject to ongoing debate. Excluding the current study, there is possibly considerable irreducible uncertainty in the provenance of the data used to date (as, for example, argued by *Marks et al., 2014*). We note another novel, independent observational process arising from the predation of foxes by wedge-tailed eagles (*Aquila audax*, Latham 1802) that should provide additional, independent inference. Like their northern hemisphere counterpart the golden eagle (*Aquila chrysaetos*, Linnaeus 1758), wedge-tailed eagles are known to effectively prey on red foxes, and red fox remains are consistently found in wedge-tailed eagle diets (at non-trivial percentages) in a wide range of habitats wherever foxes are present (e.g., *Olsen et al., 2010*; *Sharp et al., 2002*; *Parker, Hume & Boles, 2007*; *Brooker & Ridpath, 1980*; *Glen et al., 2016*). Their range encompasses all of the parts of Tasmania of interest. The nest locations of the wedge-tailed eagle pairs could be carefully searched for fox remains. Making inference from such data, particularly if no fox remains are found, will need to be conditional on an appropriate eagle-fox detection model. Factors such as the territory size of eagle pairs will set the spatial resolution of the resulting inference.

Traditional methods of inferring extinction have focussed on making inference from the sighting record (e.g., *Solow, 1993*; *Solow, 2016*), where prior positive observations are used to estimate the sighting rate/probability given species presence, and the probability of extinction is estimated accordingly given the time or then number of surveys since the last sighting (see *Caley & Barry (2014)* for a recent Bayesian implementation). The method illustrated here can make inference on the probability of extinction without the need for prior sightings from the area of interest, provided there are data on a functionally linked species available. Our approach is a logical extension of the empirical findings studies such as *Barrientos & Bolonio (2009)*, who showed that the presence of rabbits adjacent to roads increases the rate of road-kill of the European polecat (*Mustela putorius*, Linnaeus 1758). Calibrating the numerical response function is the extension that enables inference on the abundance of the predator of interest. Ideally the choice for the form of the numerical response function would be informed by previous studies in conjunction with the data. The Type III Holling numerical response used here is a more general form than, for example, the simpler Type II Holling numerical response. In our example the inference from the simpler model is near identical.

Of course, prior sightings are not required for estimating detection probabilities if the detection power of the search effort is known independently. Quantifying detection probabilities becomes difficult, however, when search effort varies in space and time, along with the wildlife species of interest. This necessitates computationally intensive methods that respect the complexity of the underlying population and surveillance processes (*Caley, Ramsey & Barry, 2015*). The key feature of the approach we have illustrated here is that by statistically conditioning on a second, biologically linked species, it essentially integrates over the unknown factors underpinning the observation effort and hence detection probability. *Reid (1995)* notes how natural the process of conditioning is as a tool in everyday statistics, and our example here demonstrates how it can be used to extract useful inference from data, for which at first glance may appear uninformative.

Finally, although we have illustrated our approach using wildlife collision data, it is applicable to any observational process which samples both the predator species of interest and a prey species (or multiple prey species), for which a numerical response relationship between the two is known to exist and can be calibrated.

## ACKNOWLEDGEMENTS

N Fulton assisted with identifying data sources. G Madden and S Godley assisted with accessing data from the Australian Transport Safety Bureau wildlife strike database. The comments of J Hone, D Parker, D Ramsey, G Smith, M Westcott and anonymous referees improved a draft manuscript.

### Funding

The work was supported by the CSIRO. The funders had no role in study design, data collection and analysis, decision to publish, or preparation of the manuscript.

## Grant Disclosures

The following grant information was disclosed by the authors:
CSIRO.

## Competing Interests

The authors declare they have no competing interests. All authors are employees of Data61, Commonwealth Scientific and Industrial Research Organisation.

## Author Contributions

- Peter Caley conceived and designed the experiments, performed the experiments, analyzed the data, wrote the paper, prepared figures and/or tables, reviewed drafts of the paper.
- Geoffrey R. Hosack performed the experiments, analyzed the data, wrote the paper, reviewed drafts of the paper.
- Simon C. Barry wrote the paper, reviewed drafts of the paper.

## Data Availability

The code has been supplied as a Supplementary File. The raw data used for analysis is contained within the text in Table 1 and as a Supplementary File.

## Supplemental Information

Supplemental information for this article can be found online at http://dx.doi.org/10.7717/peerj.3014#supplemental-information.

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
