# Peer review of "Making inference from wildlife collision data: inferring predator absence from prey strikes"

_PeerJ, doi:10.7717/peerj.3014_

## Round 0.1 · original submission · Minor Revisions

Both reviewers have only minor revisions. Please correct accordingly and I look forward to the re-submission.

·

Basic reporting

All basic reporting in the article meets the standards of PeerJ.

Experimental design

The experimental design is clearly outlined, of a high standard and appropriate for the analyses presented in the paper.

Validity of the findings

Although the data are inherently sparse (as acknowledged by the authors), the authors have done an excellent job at extracting (and aggregating where appropriate) the data and then analyzing these data in a novel way. The conclusions are appropriate and the broader application of the approach is clearly outlined.

Additional comments

I believe that this article, especially the assessment of carnivore population dynamics elsewhere in the world, is extremely important for the assessment of roadkill data globally.

There is only one small error:
Line 126: lagomorph is incorrectly spelled.

·

Basic reporting

This is an interesting and relatively novel approach to the problem of species extinction/introduction and is well written throughout.

Experimental design

This is a novel approach to looking for relationships between species occurrence and I have some minor concerns, although these do not invalidate the paper.
1. I understand that the equation used relates prey and predator numbers at a specific location. The use of 'state' may make this location rather large. Can the authors comment on this? Do foxes and lagomorphs exist in the whole of the state (in respect to the airports used) where strikes occur? Also, they have not investigated alternative models.
2. Foxes and rabbits are crepuscular. Is there any bias in time of day that these airports are used? I would expect larger airports to have relatively greater activity at dusk.
3. I found the use of airport strikes very interesting since these have to be reported. When I first glanced at the paper I was expecting to read about roadkill, which is poorly and sporadically reported. Can the authors at least comment on a lack of roadkilled foxes in Tasmania to support their findings? Alternatively one could drive roads as a transect and count roadkill, but this does not seem to be considered.

Validity of the findings

1. The Holling type III response seems to be specific to the fox/lagomorph relationship. Perhaps raise in the discussion whether alternative models would be considered in different circumstances (e.g. specifically for the eagle - fox relationship).

---

## Round 0.2 · accepted · Accept

All looks good. At production stage please put in species names on first mention in the abstract too?